

# Plant dominance in a subalpine montane meadow: biotic vs. abiotic controls of subordinate diversity within and across sites

Erika LaPlante[1,2] and Lara Souza[3]

[1] Department of Plant Biology, Michigan State University, East Lansing, MI, United States of America
[2] Department of Integrative Biology, University of Californnia Berkeley, Berkeley, CA, United States of America
[3] Oklahoma Biological Survey & Department of Microbiology and Plant Biology, The University of Oklahoma, Norman, OK, United States of America

Corresponding author
Lara Souza, lara.souza@ou.edu

## ABSTRACT

**Background**. Understanding the underlying factors that determine the relative abundance of plant species is critical to predict both biodiversity and ecosystem function. Biotic and abiotic factors can shape the distribution and the relative abundance of species across natural communities, greatly influencing local biodiversity.

**Methods**. Using a combination of an observational study and a five-year plant removal experiment we: (1) documented how plant diversity and composition of montane meadow assemblages vary along a plant dominance gradient using an observational study; (2) tracked above- and belowground functional traits of co-dominant plant species *Potentilla* and *Festuca* along a plant dominance gradient in an observational study; (3) determined whether plant species diversity and composition was directly influenced by commonly occurring species *Potentilla* and *Festuca* with the use of a randomized plot design, 5-year plant removal experiment (no removal control, *Potentilla* removed, *Festuca* removed, $n = 10$).

**Results**. We found that subordinate species diversity and compositional dissimilarity were greatest in *Potentilla* and *Festuca* co-dominated sites, where neither *Potentilla* nor *Festuca* dominated, rather than at sites where either species became dominant. Further, while above- and belowground plant functional traits varied along a dominance gradient, they did so in a way that inconsistently predicted plant species relative abundance. Also, neither variation in plant functional traits of *Festuca* and *Potentilla* nor variation in resources and conditions (such as soil nitrogen and temperature) explained our subordinate diversity patterns. Finally, neither *Potentilla* nor *Festuca* influenced subordinate diversity or composition when we directly tested for their impacts in a plant removal experiment.

**Discussion**. Taken together, patterns of subordinate diversity and composition were likely driven by abiotic factors rather than biotic interactions. As a result, the role of abiotic factors influencing local-level species interactions can be just as important as biotic interactions themselves in structuring plant communities.

## INTRODUCTION

Linkages between species' relative abundance and ecosystem function are important to predict ecosystem resistance and resilience to global change pressures. Dominant species, with high relative abundance (i.e., primary production), have been shown to strongly impact community dynamics and ecosystem function (*Whittaker, 1965*; *Wardle et al., 1999*; *Grime, 1977*; *Grime, 2001*; *Grime, 2006*; *Hooper et al., 2005*). Specifically, dominant species shape the dynamics of communities by influencing subordinate species' presence and abundance (*Garbin et al., 2016*; *Gibson et al., 2012*; *Grman & Suding, 2010*). According to *Grime*'s (*1998*) "mass ratio hypothesis" species with greater primary production exert the main controls for the functioning of ecosystems. Based on this hypothesis, dominant species are considered more important in an ecosystem because of the greater aboveground abundance of biomass or leaf area (e.g., foliar cover) that promotes resource uptake.

However, an increasing amount of recent studies (*Mariotte et al., 2013a*; *Mariotte et al., 2013b*; *Mariotte, 2014*) are showing that the subordinate species potentially have an even greater impact on ecosystem function, acting sometimes as a sieve that influences the regeneration of dominant species following perturbations (*Grime, 1998*). In fact, the importance of subordinate species may be seen during environmental change. For example, the results of a mesocosm experiment (*Kardol et al., 2010*) and a field experiment in mountain grassland (*Mariotte et al., 2013a*) demonstrated that subordinate species can produce relatively more biomass in changed environmental conditions, such as a drought, promoting community stability in a time of disturbance. This suggests that dominant species can respond strongly to the direct presence of an abiotic factor, while subordinate species can be more resistant to abiotic influences and take advantage of the reduced competition (*Mariotte et al., 2013a*). As conditions and resources vary along environmental gradients, such environmental filters can select the ability of species to acquire resources and/or tolerate conditions and dominate in local communities. In fact, dominant and subordinate species have been shown to vary in above- and belowground functional traits (e.g., specific leaf area and height) across environmental gradients (*Wellstein et al., 2013*).

Understanding the relative importance of biotic vs. abiotic processes determining the relative abundances of species (*Wisz et al., 2013*), and how these processes generate the plant diversity patterns we see across space and time is a critical ecological focus of studies. Alongside estimating species-specific abundance and biodiversity, functional traits, provide a critical link between species relative abundances and the functioning of ecosystems (*Lavorel, 2013*). Dominant and subordinate species have been shown to differ in aboveground functional traits that determine plant performance. For instance, dominant species have fast growing/high resource acquisition strategies while subordinate species are associated with resource conservation/slow growing strategies across (*Diaz et al., 2004*; *Wright et al., 2004*; *Mariotte et al., 2013b*; *Mariotte, 2014*) and within species (*Read et al., 2014*; *Korner & Renhardt, 1987*). We investigated how subordinate species diversity and composition varied along a plant dominance gradient and then we directly tested the effects of two co-occurring dominant montane meadow plant species: *Festuca thurberi* (hereafter *Festuca*) and *Potentilla graciilis* (hereafter *Potentilla*) on community

diversity and composition of subordinate species. We first conducted an observational study that compared subordinate diversity and composition along a plant dominance gradient (from *Potentilla dominance* to *Potentilla* and *Festuca* co-dominance to *Festuca* dominance). We then conducted a four-year plant removal study that directly tested the effects of *Festuca* and *Potentilla* on subordinate species diversity and compositional similarity with the following treatments: control (no plant removal), *Potentilla* removed, *Festuca* removed. Specifically, we asked the following research questions: (1) does diversity and composition of subordinate montane meadow plant assemblages vary across and within a *Festuca-Potentilla* dominance gradient?; (2) do resources and conditions differ across and within a *Festuca-Potentilla* dominance gradient?; (3) do above- and belowground functional traits differ in *Festuca and Potentilla* across the dominance gradient to explain differences in subordinate diversity and composition?

## MATERIALS AND METHODS
### Study site
Our study sites were located at the Rocky Mountain Biological Laboratory (RMBL), Gothic Colorado (latitude 38°53′N, longitude 107°02′W, elevation 2,920 m above sea level) (*Saleska, Harte & Torn, 1999*). Annual precipitation averages 750 mm, 80% of which was snow (snowmelt typically ending in May) (*Saleska, Harte & Torn, 1999*; *Harte & Shaw, 1995*). Mean daily-average summer air temperature is ∼10 °C. Mean snowfall at RMBL is 11,140 cm with a 24-year trend towards lower snowfall overtime (*Inouye et al., 2000*) with field summer seasons ranging from 0.69 m and 0.47 m (water equivalent), respectively (*Harte & Shaw, 1995*). Soil texture is a well-drained Cryoboroll, which is a deep rocky outcrop that is non-calcareous and formed on a glacial till (*Saleska, Harte & Torn, 1999*). Below a sparse litter layer (due to snowpack), the soil is uniform in color and texture down to about 50 cm. Organic content averages ∼10% at a soil depth of 5 cm below the litter layer and drops to ∼6% at 50 cm (*Harte & Shaw, 1995*). Soils at the experimental and observational sites averaged a pH of 5.7–6.3 (*Saleska, Harte & Torn, 1999*).

### Experimental design
#### Observational study
We selected three montane meadow sites based on the frequency and abundance of commonly occurring plant species: (1) Festuca dominated site: *Festuca thurberi* exhibited high abundance (i.e., *Potentilla* low abundance), (2) Co-dominated site: *Festuca* and *Potentilla* exhibited similar abundances (i.e., co-dominated), and (3) *Potentilla* dominated site: *Potentilla gracilis* exhibited high abundance (i.e., *Festuca* low abundance). Specifically, we determined plant dominance at the scale of the montane meadow site based on the relative abundance and frequency of plant species  (*Mariotte et al., 2013a*; *Mariotte et al., 2013b*; *Mariotte, 2014*); with dominant species being the most frequent and abundant (based on species-specific foliar cover measurements). In each observational site, we established 9, 1-m$^2$ plots along three parallel transects (three 1-m$^2$ plots per transect; each 1-m$^2$ plot along a transect exhibited one of the following categories: *Potentilla* Dominated, *Festuca* Dominated, and Co-Dominated by *Potentilla* and *Festuca*).

### Experimental study

We manipulated the presence of *Festuca* and *Potentilla*, which co-dominate within this existing montane meadow vegetation, across 1.5 m × 1.5 m plots ($N = 30$). The plots were spaced one meter from each other in a completely randomized plot design with the following three treatments: (1) control (no plant species removed), *Festuca* removed, *Potentilla* removed. In removal treatments, plant species were clipped (to one cm from the ground) every other week throughout the growing season (June–August), for three growing seasons (2013–2015).

## Plant community measurements

To examine how subordinate diversity varied along a plant dominance gradient (*Observational Study*) and how dominant species directly affected subordinate diversity (*Experimental Study*), we measured species-specific foliar cover, species richness (the number of species), Shannon's diversity and evenness in each observational and experimental plot twice in each growing season (Observational Study = one growing season, Experimental Study = three growing seasons). To estimate species-specific foliar cover, we used a modified Braun-Blanquet scale that included six categories: <1%, 1–5%, 5–25%, 25–50%, 50–75%, 75–100%. The median of each cover class category was assigned to each plant species in each plot and used as an estimate of species-specific abundance. Shannon's diversity ($H'$) was calculated as: $H' = -sum(pi * (ln * pi))$ and evenness was calculated as $J' = H'/S$, $S$ is species' richness.

## Above and below ground functional trait measurements

To determine above- and belowground variability across the three sites (varying in dominance of *Festuca* and *Potentilla*), two 30-meter transects were established at each site (outside of our plant sampling plots). Every six meters, a 1 m × 1 m plot was placed (totaling 5, 1-m$^2$ sampling plots per transect and $N = 10$ per site). At each plot for *Festuca* and *Potentilla*, percent species-specific foliar cover was recorded (see methods above) and leaves and roots were harvested according to the methods by *Cornelissen et al. (2003)*. We quantified specific leaf area (SLA) by harvesting three relatively young, but fully expanded leaves from each individual. Leaves were then scanned and we used ImageJ to estimate leaf area (cm$^2$). Leaves were then oven dried for approximately 48 hours at 65 °C and weighted. We then divided area by mass to obtain SLA (cm$^2$ g$^{-1}$). We sampled absorptive roots from a single individual of each species (*Festuca* and *Potentilla*) per plot in each transect to estimate specific root length (SRL, cm g$^{-1}$). For *Potentilla* we dug up the entire plant and root systems with a spade and then bagged the entire mass for later analysis. For *Festuca* we used a soil core (one cm in diameter ×10 cm in length) to sample roots from the plant species, by angling the soil core into the base of the plant, to ensure only roots of that species were extracted. Ten fine root pieces from each core were separated and used for analysis. Using ImageJ again, we scanned and interpreted the data using a Plugin called 'IJ Rhizo v0beta'. We then oven dried roots for approximately 48 hours at 65 °C and weighted.

## Microclimate measurements

To determine how resources and conditions varied along a dominance gradient (*Observational study*), as well as being impacted by dominant species (*Experimental study*), we tracked light and soil nutrient availability (resources) as well as soil temperature (conditions). We measured photosynthetic active radiation (PAR, hereafter light availability) once during the peak of the growing season (July) in each of the experimental and observational plots. Soil temperature and soil nitrogen availability were measured overtime by being deployed, ibuttons and resin bags, in early June and retrieved from plots in late July. To estimate light availability below the vegetation canopy, we used a line-integrating ceptometer (Decagon Accupar; Decagon Devices, Pullman, WA) with all light availability measurements made on clear days between 11 am and 2 pm. Specifically, we placed the line-integrating ceptometer approximately 2 centimeters from the ground. To determine soil temperature, we used ibuttons (MAXIM) that recorded surface soil temperature every minute. To assess the availability of NO3-N and NH4-N in the soil solution, we placed mixed-bed ion-exchange resin bags in nylon stockings (H-OH form, # R231-500; Fisher Scientific International Inc., Pittsburgh, PA, USA) at five-cm soil depth at two locations in each of the experimental and observational plots (*Hart et al., 1994*). Resins were then air-dried, and two g of resins from each plot were extracted with 2 M KCl. Pool sizes of NO3_ and NH4 + were analyzed on a Lachat AE Flow Injection Autoanalyzer (Lachat Quikchem 8000; Hach Corporation, Loveland, OH). All values expressed in this article are based on air-dried resins.

## Statistical analyses

To determine how subordinate species diversity, as well as above- (SLA) and below-ground (SRL) plant functional traits, varied along a plant dominance gradient (*Observational study*), we ran a series of one-way analyses of variance (ANOVAs) with 'site' as our main factor (e.g., *Potentilla* dominated, *Festuca* dominated, Co-dominated). To determine the direct role of dominant species on subordinate species diversity we performed one-way ANOVAs with 'plant removal' (control, *Potentilla* removed, *Festuca* removed) as our main fixed effect (*Experimental study*). All the ANOVA analyses were conducted using Jump 11 (JMP).

To determine (1) how compositional similarity of subordinate species varied along a plant dominance gradient and (2) how dominant species affected compositional similarity of subordinate species, we generated a Bray-Curtis similarity matrix from the log transformed plant composition (log x+1). We then performed a permutational multivariate analysis of variance (PERMANOVA; *Anderson, 2001*) on the Bray Curtis similarity matrix. A pseudo F-ratio is calculated within the PERMANOVA framework comparing the variability in species composition both *within* treatments and *among* treatments based on the observed variability in species composition vs. the variability in species composition using a generated null distribution (*Anderson, Ellingsen & McArdle, 2006*). PERMDISP (permutational multivariate analysis of dispersion) analysis, on the other hand, is a measure of 'dispersion' of community composition in multivariate space (*Anderson, Ellingsen & McArdle, 2006*). We used PRIMER version 1.0.3 (Plymouth Marine

Laboratory, UK) for these analyses. We performed a series of principal coordinate analyses (PCoA) to illustrate species compositional similarity and dissimilarity in a two-dimensional multivariate space,. Finally, we used a similarity percentage analysis (SIMPER) to determine the relative contribution of plant species driving compositional dissimilarities both in the observational as in the experimental study.

# RESULTS

## Community structure and compositional similarity across a dominance gradient (Observational study)

Plant community structure differed across a dominance gradient. Co-dominated sites, generally, had greater total cover, richness, evenness and diversity than *Potentilla* or *Festuca* dominated sites (Table 1, Fig. 1A). For example, total cover was 26% greater while evenness, richness, and diversity were 6%, 24%, and 15% greater respectively when both *Festuca* and *Potentilla* co-dominated than when either species became dominant. Plant species composition, similar to community diversity, differed across a dominance gradient (Table 1, Fig. 1B). While all sites differed from one another in compositional similarity, co-dominated sites differed the most in compositional similarity to either *Potentilla* or *Festuca* dominated sites.

## Shifts in above-and belowground functional traits across a dominance gradient (Observational study)

Both above- and belowground plant functional traits varied along a plant dominance gradient, but plant identity dictated such variation. For example, the average area of *Festuca* leaves ($F = 9.24$, $P = 0.001$), but not SLA ($P > 0.05$), was 40% greater in *Festuca* dominated ($9.79 \pm 0.54$) and co-dominated sites ($11.15 \pm 0.75$) than in *Potentilla* dominated sites ($6.86 \pm 0.84$) where *Festuca* is subordinate. On the other hand, *Festuca* SRL was 38% greater when it became subordinate (e.g., *Festuca* $2712.88 \pm 351.58$ in *Potentilla* dominated site) than when it dominated local communities ($1990.39 \pm 214.44$ in *Festuca* dominated site) ($F = 9.37$, $P = 0.001$).

Similarly, *Potentilla* differed marginally in aboveground functional traits, while differing strongly in belowground functional traits across a plant dominance gradient. For the aboveground functional traits, the average area of *Potentilla* leaves was greater in *Festuca* dominated than co-dominated or *Potentilla* dominated sites ($F = 2.83$, $P = 0.07$). *Potentilla* dominated site leaf area was on average $143.9 \text{ cm}^2 \pm 19.7$ while in co-dominated site and *Festuca* dominated site leaf area was $171.3 \text{ cm}^2 \pm 6.9$ and $170.2 \text{ cm}^2 \pm 20.35$, respectively. Specific leaf area, on the other hand, did not differ ($F = 0.84$, $P = 0.44$) across dominance gradient for *Potentilla* (*Potentilla* dominated site: $103.4 \text{ cm}^2 \pm 10.8$; Co-dominated site: $102.5 \text{ cm}^2 \pm 10.21$; *Festuca* dominated site: $131.72 \text{ cm}^2 \pm 8.42$). Belowground functional trait (SRL) for *Potentilla* was 30% greater when ($F = 3.77$, $P = 0.35$) *Potentilla* was a co-dominant than when it was a dominant (*Potentilla* dominated site) or subordinate (*Festuca* dominated site).

**Table 1 Plant dominance and subordinate diversity and composition across sites.** Results from plant dominance effects based on (A) a one-way analysis of variance (ANOVA) testing for effects on total cover, richness, evenness and diversity including *F*-ratio (*F*) and *P*- values (*P*) across time (June and July). (B) A multivariate permutation analysis of variance (PERMANOVA) testing for effects of plant dominance on subordinate species' composition including Pseudo-*F* and *P* (perm) across time (June and July).

**(A)**

| Source | June | | July | |
|---|---|---|---|---|
| | *F* | *P* | *F* | *P* |
| **Total cover** | | | | |
| Dominance | 9.9815 | **0.0006** | 6.9543 | **0.0037** |
| **α diversity** | | | | |
| Dominance | 1.1713 | 0.3252 | 5.2 | **0.0123** |
| **Shannon's evenness** | | | | |
| Dominance | 2.2157 | 0.1284 | 7.31 | **0.0029** |
| **Shannon's diversity** | | | | |
| Dominance | 7.8929 | **0.002** | 11.845 | **0.0002** |

**(B)**

| Source | *df* | SS | MS | Pseudo-*F* | *P*(perm) |
|---|---|---|---|---|---|
| **June- across site** | | | | | |
| Dominance | 2 | 20,491 | 10,245 | 14.838 | **0.0001** |
| Residual | 45 | 31,071 | 690.47 | | |
| Total | 47 | 51,562 | | | |
| **July- across site** | | | | | |
| Dominance | 2 | 16,977 | 8,488.5 | 12.383 | **0.0001** |
| Residual | 45 | 30,848 | 685.51 | | |
| Total | 47 | 47,825 | | | |

## Community structure, compositional similarity (experimental study)

We found that neither dominant species, *Potentilla* or *Festuca*, affected plant richness, evenness and diversity (Table 2, Figs. 2A–2D, Appendix Fig. S1). Similarly, compositional similarity was not impacted by the removal of neither dominant species (Table 2, Figs. 2E–2F, Appendix Fig. S2).

## Microclimate across a plant dominance gradient (Observational & Experimental study)

We found light availability and temperature, but not soil N availability, to vary along a plant dominance gradient. Light availability was 25% greater in *Festuca* dominated sites than co-dominated or *Potentilla* dominated communities. Further, co-Dominated sites and *Festuca* dominated sites had the largest minimum and maximum temperature difference (60.20 °C and 60.11 °C, respectively). *Potentilla* dominated sites had a lower temperature difference of 53.28 °C, which coincides with having the lowest light availability measurements within these plots. Finally, we found no effects of dominant species on resources (light availability and soil N) or conditions (soil temperature) (Table 3).

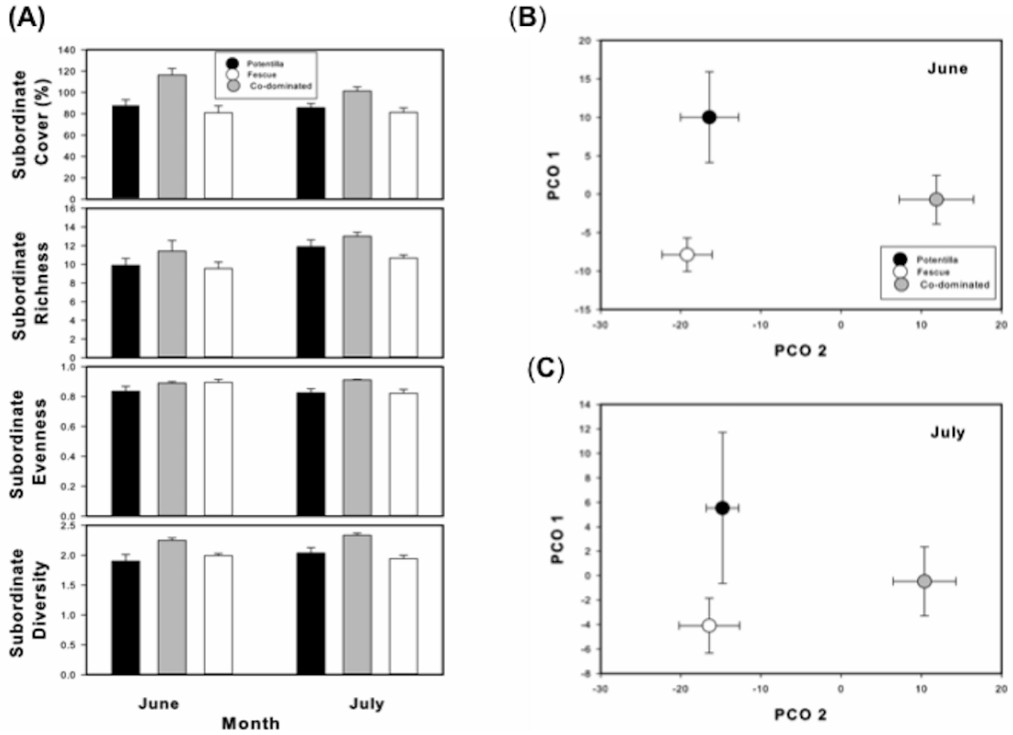

**Figure 1 Subordinate diversity and composition across a plant dominance gradient.** (A) Mean (±standard error) subordinate species' cover, richness, evenness and diversity across a plant dominance gradient (*Potentilla* dominated, *Potentilla* and *Festuca* co-dominated, *Festuca* dominated) for June and July of 2014. (B) Principal Coordinate Ordination (PCO) illustrating in a two-dimensional scale (PCO Axis 1 and PCO Axis 2) subordinate species composition across a plant dominance gradient (*Potentilla* dominated, *Potentilla* and *Festuca* co-dominated, *Festuca* dominated) (observational study) for June and (C) July of 2014.

## DISCUSSION

Subordinate species diversity and composition varied along a dominance gradient with highest diversity, yet lowest compositional similarity, in plant communities co-dominated by both *Festuca* and *Potentilla*, rather than communities dominated by either species. In other words, in co-dominated sites, where *Potentilla* and *Festuca* were equally abundant, subordinate diversity and compositional dissimilarity were the greatest. While above- and belowground plant functional traits varied along a dominance gradient, neither above- nor belowground plant functional trait of *Festuca* and *Potentilla* consistently predicted relative abundance. Further, variation in resources and conditions did not explain our subordinate diversity patterns. Taken together, patterns of subordinate diversity and composition across meadow sites are likely driven by biotic interactions and abiotic factors unaccounted for in our measurements. Finally, neither *Potentilla* nor *Festuca* short-term removal influenced subordinate diversity or composition when we directly tested for their impacts in a plant removal experiment. Plant removal effects, particularly on belowground structure and function, can be buffered temporally.

**Table 2  Plant removal effects on subordinate diversity and composition.** Results from plant removal effects based on (A) a one-way ANOVA testing for effects on total cover, richness, evenness and diversity including $F$-ratio ($F$) and $P$- values ($P$) across time (June and July). (B) PERMANOVA testing for effects of plant removal on subordinate species' composition including Pseudo-$F$ and $P$ (perm) across time (June and July).

**(A)**

| | Co-dominated | | | |
| | June | | July | |
| Source | $F$ | $P$ | $F$ | $P$ |
|---|---|---|---|---|
| **Total cover** | | | | |
| Dominant removal | 0.01 | 0.98 | 4.16 | **0.03** |
| **$\alpha$ diversity** | | | | |
| Dominant removal | 0.06 | 0.94 | 0.83 | 0.44 |
| **Shannon's evenness** | | | | |
| Dominant removal | 1.07 | 0.36 | 0.16 | 0.85 |
| **Shannon's diversity** | | | | |
| Dominant removal | 0.21 | 0.81 | 0.61 | 0.54 |

**(B)**

| Source | df | SS | MS | Pseudo-$F$ | $P$(perm) |
|---|---|---|---|---|---|
| **Co-dominated site june** | | | | | |
| Removal | 2 | 1,733.1 | 866.57 | 1.3565 | 0.1682 |
| Residual | 27 | 17,249 | 638.84 | | |
| Total | 29 | 18,982 | | | |
| **Co-dominated site july** | | | | | |
| Removal | 2 | 1,022.1 | 511.06 | 0.70289 | 0.8319 |
| Residual | 27 | 19,631 | 727.09 | | |
| Total | 29 | 20,654 | | | |

## Community diversity and compositional similarity across a dominance gradient: observation vs. experiment

Co-dominance by *Potentilla* and *Festuca* was associated with greater subordinate species' abundance and overall diversity than when either *Potentilla* or *Festuca* were dominant in a plot, along a dominance gradient. Dominant species have been shown to strongly impact subordinate species' abundance and biodiversity by disproportionately utilizing resources or conditions that would otherwise be available for subordinate species, especially in favorable (*Wardle et al., 1999*; *Wilsey & Polley, 2002*; *Diaz et al., 2003*) rather than unfavorable environments (*Smith et al., 2004*). Under favorable conditions, dominant species can have antagonistic effects on subordinate counterparts given their higher competitive abilities with higher resource availability. Over–yielding, dominant individuals may modify resources and conditions for subordinates drastically. Similar to our documented patterns, *Suding & Goldberg (2001)* found dominant species to reduce subordinate biodiversity by monopolizing resources and therefore exhibiting greater resource uptake rates, reducing subordinate species' abundance. Plant dominance may not only lower biodiversity at the plot level, but overall diversity among subordinate
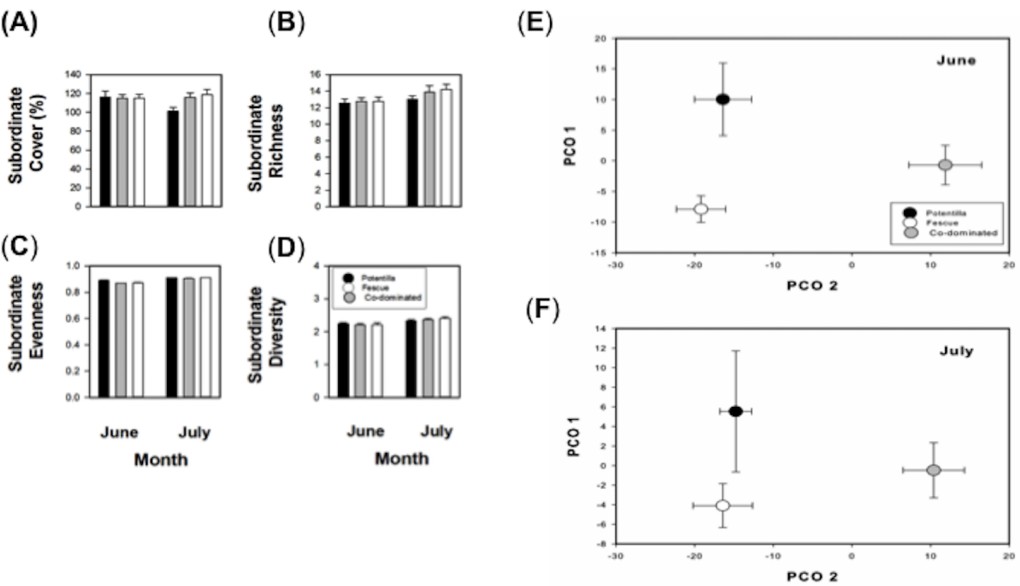

**Figure 2** **Plant removal effects on subordinate diversity and composition.** Mean (±standard error) (A) subordinate species' cover, (B) richness, (C) evenness and (D) diversity across plant removal treatments (*Potentilla* removal, *Festuca* removal, no removal control) for June and July of 2014. (E) PCO illustrating in a two-dimensional scale (PCO Axis 1 and PCO Axis 2) subordinate species composition across plant removal treatments (*Potentilla* removal, *Festuca* removal, No plant removal) (Experimental Study) for June and (F) July of 2014.

assemblages. Belowground biotic interactions are also critical in determining dominance patterns and interactions between dominant and subordinate species. For instance, fungi and earthworms have been shown to strongly influence the individual growth but also the composition of subordinate species in ecological communities (*Mariotte et al., 2015*; *Mariotte et al., 2016*; *Mariotte, Canarini & Dijkstra, 2017*). Further, lower among assemblage diversity or spatial–temporal homogenization of subordinate species, can lead to a potentially long-term biodiversity deficit due to a lower regional species-pool which won't be resupplying local subordinate assemblages with more plant species propagules (*Huston, 1999*; *Witman, Etter & Smith, 2004*). In other words, co-dominance patterns could promote short- (richness, evenness, diversity) and long-term (compositional similarity) biodiversity patterns in montane meadows.

However, when we directly tested for the effects of co-dominant species to influence subordinate biodiversity we found that removing either *Festuca* or *Potentilla* did not affect subordinate diversity. Similar to our findings, *Smith & Knapp (2003)* also found that dominant species did not affect subordinate species diversity. Smith and Knapp found that after removing dominant $C_4$ grasses the subordinate assemblage in the grassland ecosystem did not compensate for the loss of dominant species. Instead, they found that subordinate productivity was unaffected by even a 50% reduction in density. In a field experiment conducted by *Souza, Weltzin & Sanders (2011)*, diversity of the subordinate community was found to be on average 20% greater in plots with the removal of a dominant

**Table 3 Resources and conditions influenced by dominant plant species.** ANOVA table indicating the mean and (SE) of (A) light availability (PAR), minimum and maximum temperature, along with model $F$-ratio and $P$-value and (B) soil ammonium, soil nitrate, and total soil nitrogen, along with model $F$-ratio and $P$-value for observational and experimental studies.

**(A)**

| | Ammonium | | | | Nitrate | | | | Total Nitrogen | | | |
|---|---|---|---|---|---|---|---|---|---|---|---|---|
| | Mean | SE | *F* | *P* | Mean | SE | *F* | *P* | Mean | SE | *F* | *P* |
| **Observational Study** | | | | | | | | | | | | |
| *Potentilla* dominated | 2.18 | 0.64 | 1.62 | 0.22 | 0.28 | <0.01 | 0.28 | 0.75 | 2.47 | 0.67 | 1.62 | 0.22 |
| Co-dominated | 0.98 | 0.57 | | | 0.30 | <0.01 | | | 2.88 | 0.58 | | |
| *Festuca* dominated | 2.42 | 0.67 | | | 0.29 | <0.01 | | | 4.77 | 0.63 | | |
| **Plant removal experiment** | | | | | | | | | | | | |
| Control | 0.98 | 0.22 | 0.72 | 0.49 | 0.29 | 0.01 | 1.19 | 0.31 | 1.27 | 0.47 | 0.72 | 0.49 |
| *Potentilla* removal | 1.67 | 0.77 | | | 0.29 | 0.01 | | | 1.96 | 0.55 | | |
| *Festuca* removal | 1.75 | 0.57 | | | 0.30 | 0.01 | | | 2.04 | 0.55 | | |

**(B)**

| | PAR (umol photons m$^{-2}$s$^{-1}$) | | | | Min temperature (°C) | | | | Maximum temperature (°C) | | | |
|---|---|---|---|---|---|---|---|---|---|---|---|---|
| | Mean | SE | *F* | *P* | Mean | SE | *F* | *P* | Mean | SE | *F* | *P* |
| **Observational study** | | | | | | | | | | | | |
| *Potentilla* dominated | 970.1 | 83.6 | 6.37 | 0.01 | 3.78 | 0.26 | 3.25 | 0.06 | 57.06 | 2.89 | 1.26 | 0.31 |
| Co-dominated | 976.7 | 47.6 | | | 3 | 0.52 | | | 63.2 | 4.11 | | |
| *Festuca* dominated | 1,310.6 | 99.3 | | | 2.44 | 0.44 | | | 62.56 | 2.64 | | |
| **Experimental study** | | | | | | | | | | | | |
| *Festuca* removal | 1,089.2 | 261.50 | 2.36 | 0.11 | 3.17 | 0.17 | 0.61 | 0.56 | 56.83 | 7.91 | 1.38 | 0.29 |
| Control removal | 976.7 | 47.60 | | | 3.00 | 0.52 | | | 63.20 | 4.11 | | |
| *Potentilla* removal | 1,086.6 | 47.20 | | | 2.30 | 0.34 | | | 68.10 | 2.50 | | |

forb species, *Solidago altissima.* Similarly, in *Verbesina* removal plots, diversity on average was 30% greater than in plots where *Verbesina* was present. Even though the removal of dominant species affected diversity and evenness, there were no effects on composition of these plots, because richness did not change (*Souza, Weltzin & Sanders, 2011*). On the other hand, dominant species may attain high abundance by being good 'stress-tolerators' rather than a great competitor relative to other species (*Read et al., 2017*) that have low abundance and classified as subordinate or transient.

*Whittaker (1965)* suggests that a closer look should be taken to differentiate between subordinate and transient species, as there is a keen distinction that separates them. Where subordinates consistently co-occur with specific dominants in larger abundance than the dominants, though smaller in build, transient species lack consistency of association with dominants and infrequent occurrence temporally and spatially. Transient species have been found to make a small contribution to biomass, though most are species that occur as dominants or subordinates in other communities, often nearby. Though through our observations and experiments, we found that dominant species did not affect subordinate diversity, as the majority of the plant species in our montane meadows were transient and very few were actually subordinate species (see Appendix Table S1).

Changes in biodiversity, along the plant dominance gradient, translated into divergence in subordinate species' compositional similarity. Co-dominated subordinate communities exhibited greater equitability of subordinate forbs than either *Festuca* or *Potentilla* dominated communities that exhibited two main subdominant forbs making up subordinate species assemblages. Co-dominated communities had a greater proportion of perennial forb species that differed in identity from perennial forbs in sites domintated by either *Potentilla* or *Festuca*. For example, co-dominated communities had a greater abundance of *Erigeron speciosa*, *Artemesia ludiciviana*, and *Fragaria virginiana* than *Potentilla* or *Festuca* dominated sites (which had greater proportion of perennial forbs such as *Helianthella quinquenervis* and *Thalictrum fendleri*) which is a clear shift in composition.

Similar to our documented patterns, dominant species have been found to alter compositional similarity of subordinate assemblages (*Grime, 1998*). However, when we directly tested for the effects of co-dominant species to influence subordinate species composition we found that removing either *Festuca* or *Potentilla* did not affect subordinate compositional similarity. Similar to our findings, *Souza, Weltzin & Sanders (2011)* also found that dominant species did not affect subordinate species composition in an old-field ecosystem. For instance, when *Souza, Weltzin & Sanders (2011)* removed either $C_3$ perennial forb: *Solidago* or *Verbesina* species, compositional similarity of subordinate species did not converge or diverge relative to dominant species removal treatments. Further, other plant removal studies were only able to detect strong effects of dominant species on subordinate species composition over longer time frames (e.g., 8 years or longer) (*Schmitz, 2003*; *Munson & Lauenroth, 2009*).

## Shifts in above-and belowground functional traits across a dominance gradient

Plant functional traits varied along the plant dominance gradient, but the documented patterns did not support our original prediction that functional traits would be associated with dominance patterns. Specific root length, plant allocation towards greater investment on root area than mass increasing surface area to volume ratio that promotes greater resource uptake, increased for both *Festuca* and *Potentilla* when they became more subdominant than dominant. Such shift in belowground traits for both *Festuca* and *Potentilla* likely resulted from greater resource competition when they are subdominant than dominant. Such belowground strategy differs from other studies that have found subordinate species to generally have root traits associated with resource conservation rather than rapid acquisition (*Mariotte et al., 2013a*; *Mariotte, 2014*). Perhaps montane plant communities with narrower growing season windows relative to other systems, foster greater plasticity in belowground traits that promote persistence of subdominant species. Surprisingly aboveground functional traits, such as SLA did not shift when *Festuca* and *Potentilla* became subdominant. Greater total leaf area production (e.g., greater foliar cover) in dominated sites promoted dominance regardless of changes in leaf function. There are many different factors that can contribute to a lack of correlation in diversity and above- and belowground functional traits, such as abiotic constraints (*Hooper et al., 2000*): species or groups of plants could be responding to different abiotic constraints, such as

soil nutrients and water availability (*Hooper et al., 2000*). Though above- and belowground functional traits do not directly associate with species relative abundance, functional traits of dominant plant species influence ecosystem resilience and resistance. Generally, communities dominated by slow growth plants tend to have low resilience and high resistance, while the opposite is true for communities dominated by fast growing plants (*Aerts, 1995*; *Leps, Osbornovakosinova & M, 1982*; *Macgillivray et al., 1995*). However, a recent study performed in a montane meadow nearby (with greater dominance of *Festuca* than our plant removal plots) found fast compensatory responses of functional traits in subordinate species in removal relative to control plots (*Read et al., 2017*).

## Biotic and Abiotic filters determining species' relative abundances

Biotic and abiotic filters can determine the distribution and relative abundances of species across space and time. Abiotic filters, such as environmental factors like climate, can dictate the distribution and relative abundance of species across biomes (*Whittaker, 1965*; *Grime, 1979*; *Huston, 1999*; *Pavoine et al., 2011*). Biotic filters, such as species interactions as in the form of predation or competition, can influence the relative abundance of species in local assemblages (*Mouquet & Loreau, 2003*). Subordinate diversity and composition in our system are likely shaped by differences in environmental factors. Similarly, sedge dominated plots varied in relative abundance due to soil nutrient as an abiotic factor in montane meadows studied by *Theodose & Bowman (1997)*. In these montane dry meadow and wet meadow sites, Theodose and Bowman observed changes in community composition and diversity following additions of nitrogen and phosphorous fertilizers over a five-year study. In the dry meadow, Theodose and Bowman found species diversity increased significantly with fertilization, in the form of Nitrogen and Phosphorus, over the course of the study. This increase of diversity seems to have been due to an increase in the relative abundance of rare species, while the dominant species declined. In juxtaposition, the wet-meadow species diversity decreased in response to fertilization over the course of the study. This comparison allows for the comparison of the effects of fertilization on diversity between communities that differ in resource availability (*Theodose & Bowman, 1997*).

## CONCLUSIONS

Our study asked: (1) does diversity and composition of subordinate montane meadow plant assemblages vary across and within a *Festuca-Potentilla* dominance gradient?; (2) do resources and conditions differ across and within a *Festuca-Potentilla* dominance gradient?; (3) do above- and belowground functional traits differ in *Festuca and Potentilla* across the dominance gradient to explain differences in subordinate diversity and composition? We found that subordinate species diversity varies along a plant dominance gradient, peaking when both dominant species co-dominated. We also found that above- and belowground functional traits varied along a plant dominance gradient, but not in always in a predictable way of species' relative abundances. In other words, above- and belowground plant functional traits of dominant species did not consistently exhibit highest values at high relative abundance and low at lower abundance. Having said that, we only measured two functional traits and expanding such measurements could elucidate the role of above- and

belowground functional traits in our system. Finally, co-occurring dominant species did not influence the diversity or compositional similarity demonstrated in our short-term 3-year plant removal experiment. Together, abiotic factor and biotic interactions likely shape dominance patterns and subordinate diversity and composition in *Festuca* and *Potentilla* dominated montane meadow communities; with the direct and indirect effects of dominant species on co-occurring subordinates taking place over several growing seasons.

## ACKNOWLEDGEMENTS

We thank Quentin Read and Katharine Stuble for helpful comments when planning this experiment. Karissa Dunbar, William Farrell, Cindy Jatul, Helen Thayer contributed to the field and lab work associated with this experiment.

### Funding
The field component of the study was financially supported by University of Oklahoma start up funds to Lara Souza. The funders had no role in study design, data collection and analysis, decision to publish, or preparation of the manuscript.

### Grant Disclosures
The following grant information was disclosed by the authors:
University of Oklahoma.

### Competing Interests
The authors declare there are no competing interests.

### Author Contributions

- Erika LaPlante conceived and designed the experiments, performed the experiments, contributed reagents/materials/analysis tools, prepared figures and/or tables, authored or reviewed drafts of the paper, approved the final draft.
- Lara Souza conceived and designed the experiments, performed the experiments, analyzed the data, contributed reagents/materials/analysis tools, prepared figures and/or tables, authored or reviewed drafts of the paper, approved the final draft.

### Data Availability
The raw data are provided in a Supplemental File.

### Supplemental Information
Supplemental information for this article can be found online at http://dx.doi.org/10.7717/peerj.5619#supplemental-information.

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
