# Peer review of "Plant dominance in a subalpine montane meadow: biotic vs. abiotic controls of subordinate diversity within and across sites"

_PeerJ, doi:10.7717/peerj.5619_

## Round 0.1 · original submission · Major Revisions

In this study the authors uses both observational and experimental studies to address what controls subordinate diversity and composition. The manuscript has been reviewed by 2 referees and both point out that the experimental design needs to be clarified. I agree and for instance how long did the removal experiment ran? In the Abstract you say 5-years, in the Introduction 4-years and in Material and method it is 3-years. You also need to consider in depth the validity of your findings and the referee’s suggestions.

·

Basic reporting

The manuscript is well-written and well-structured. I found that the literature related to how dominant species do or do not control subordinate species composition, abundance or presence was very limited in the introduction even though this is one of the tested hypothesis in this manuscript (see for example, Garbin et al. 2016 PPEES, Gibson et al. 2012 Oikos, Grman and Suding 2010 Rest. Ecol.)

Experimental design

The research fits well with the scope of PEERJ and working hypothesis and questions are well defined in the introduction. The observational study design is however unclear. I understand that 2 meadow sites where chosen, one site with dominance of Festuca, one site with dominance of Potentilla and one site with co-dominance between both species. Then 9 plots of 1 m2 are placed in each site using 3 transects per site with 3 plots per transect. However I don't understand (line 150-153) why among each transect inside each site, there is one plot with Potentilla dominated, one plot with Festuca dominated and one co-dominated. Is dominance (Festuca or Potentilla) determined along the three meadow site? or within each site? It is very quite confusing.

There is a lack of explanation about how you define dominant and subordinate species. When reading the text, it seems that subordinate species are all species that are not Festuca or Potentilla (and there are no transients) - graph suggests the same. Then looking at Appendix Table 1, it seems that that you also classified species based on my paper (Mariotte 2014) but it is not mentioned in the text. Appendix Table 1 shows only one species dataset but you used 3 meadow sites, are species the same in the three sites? I think that the species selection and design should be more detailed/explained because it is very confusing in the current version of the manuscript.

In term of results, I would suggest to better use the data (abiotic factors) of light availability, temperature, Soil N availability etc, maybe in a RDA of the subordinate species composition constrained by dominance type (Festuca, Potentilla, Co-dom) and all these abiotic factors you measured. An option would be to transform the treatment (dominated, removal, control) into a continuous variable (relative abundance of Festuca and relative abundance of Potentilla that the RDA could be constrained by in addition to all abiotic factors. Might be better than Table 5 and 6 but just a suggestion.

Validity of the findings

Overall the discussion and conclusion are well-written and sound but this can only be adequately judged after some clarifications are made to the method section (see my previous comments).

I would tone down the idea that subordinate species diversity and abundance patterns are not controlled by biotic factors but rather by abiotic factors because you forgot about biotic interactions with soil organisms (fungi, AMF, earthworms) which has already been shown to strongly impact the growth, drought resistance, and composition of subordinate species (Mariotte et al. 2015 Func Ecol, Mariotte et al. 2016 SBB, Mariotte et al. 2017 J Ecol). It is worth discussing this topic in addition to the role of abiotic factors.

Additional comments

Overall, I obviously like the idea of the manuscript which is in line with my own research during the past 10 years. I would recommend to clarify the method section about the plant species selection and the observational study, and also take into account of my other suggestions.

Reviewer 2 ·

Basic reporting

English and writing style in this manuscript need to be improved. Many mistakes in the text. (see my specific comments below)
Literature references of this manuscript are not sufficient. There are many removal studies that can provide possible explanations for the observed patterns, while the authors only referred to about five studies.
Lacks a figure or table for the results of functional trait measurements. Table 1 ang Fig. 1, Table 3 and Fig. 3 can be combined to make the manuscript more concise. Legends of the figures should be more explicit.

Experimental design

Research questions are not well defined. In the introduction the authors did not provide solid background why plant functional traits are relevant in this study. The author only measured two traits, SLA and SRL, which I think is not enough to cover the trait diversity for a meadow ecosystem, unless the authors can provide evidence that in this specific ecosystem the two traits are the main source of trait variation. In addition, the authors only measured the traits in the observational study only for the two dominant species. As the focus of this manuscript seems to be on subordinate species, I do not see the logic for not measuring traits of subordinate species. Also, the authors only reported cover/abundance data for subordinate species, but I think the abundance of dominate species is also important to consider, as removal of one co-dominant species may cause over-production of another co-dominant species rather than subordinate species. Also, in the co-dominance plots, sub-ordinate species probably also suffered from the overall dominance by the two species.
The study sites selected, and the methods used are not sufficiently described.

Validity of the findings

Conclusions are not well supported by available results. The authors claim that subordinate species diversity and composition were likely driven by abiotic factors rather than biotic interactions, however, in the results subordinate diversity and composition were significantly different among communities with different dominant species, suggesting that dominant species had large effects on subordinate species, either directly through competition or indirectly through changes in abiotic conditions. The authors did not provide information about species pools at the study site for these different communities, and measurements on abiotic factors, plant traits and plant interactions are limited, making it hard to attribute the observed patterns to any factors.

Additional comments

Specific comments
The title is too large and vague. When I first saw this title I was imagining a study based on global data.
The introduction is too long and some parts are redundant. The logic of the introduction is not very clear. It is good to introduce the importance of subordinate species in the first paragraph but it is too long as it is now. The second paragraph introducing plant functional traits is very abrupt, and there is no connection with the first paragraph. Of course functional traits of dominant species can influence performance of subordinate species, but why can they indicate species abundance and diversity? The third paragraph can be shortened too. In addition, hypothesis 1 and 3 are the same: dominant species alter subordinate diversity and composition, so they vary along a plant dominance gradient.
L63: remove ‘In fact’. It is a hypothesis, not a fact.
L73: what does ‘which’ refer to?
L81–84: maybe just simply say that environmental conditions can shape community structure.
L85: ‘trait attributes’ to ‘traits’
L86: add references.
L89–91: it is repetition to the last sentence of the first paragraph.
L95: add references.
L100–101: why quote marks for aboveground and belowground?
L110: remove ‘For example’
L114: ‘to influence’ to ‘on’
L132: what period are the mean temperature and precipitation for?
L143: add the criteria for the selection of different site type. For example, based on relative coverage of the two dominant species?
L148: do the authors mean beginning of June to late July?
L150–153: this sentence is very confusing. Earlier in this paragraph it is introduced that each site had 3 transects, so 9 transects in all. However here as I understand each transect passed through all 3 sites. Please clarify.
L160: how frequently or how many times were the plants removed?
L164: add ‘the’ in front of ‘three sites’
L164–165: remove ‘(ranging in …)’
L167–168, 184: how was percent coverage recorded? Please introduce briefly the methods of Cornelissen et al (2003)
L175: how large was the soil core?
L188: what does ‘S’ refer to in the formula?
L191: when were the microclimate measurement made? Were the values three year average?
L198: was the PAR measurement made below the dominant canopy?
L204: when were the resin bags harvested and how long had they been in the soil?
L212, 215: it may be confusing to use ‘fixed effect’ here, as readers may think there are random effects too.
L229: ‘PCO’ to ‘PCoA’
L231: remove one ‘a’
L242: ‘similarly’ to ‘similar’
L247: ‘dominated sites’ to ‘sites dominated’
L247–251: maybe move to discussion
L264–265: if I understand right, it should be ‘… greater in Festuca dominated and co-dominated than in Potentilla …’
L275, 277: should be ‘the removal of neither dominant species’
L307–308: remove ‘manipulated to become’, as this is about the observational study and it was not manipulated
L319: add references
L349: remove ‘as expected’. If that dominant species does not affect subordinate diversity was what you prior expectation, why would you do this experiment?
L387: ‘low resistance’ should be ‘high resistance’
Table 1 and 3: does alpha diversity refer to species richness?
Table 5: why are some SEs larger than means, which suggests there are negative values?
Fig 1–4: please clarify in the legends whether they are Potentilla/Festuca dominated or removal.

---

## Round 0.2 · Minor Revisions

We have had one reviewer on this revision who, as I, thinks you have adequately answered the reviewers’ questions and integrated their comments. There are though some minor revisions needed and except from the reviewer’s suggestion I also have some minor edits that need to be addressed.
Line 70: Please give references to the “recent studies” mentioned
Lines 105-107: I think that a word is missing in research question 1?
Lines 415-416: The same here, a word seems to be missing.

·

Basic reporting

In figure 2 I believe that the legend in graph (D) should be Potentilla removal, Festuca removal and No removal (instead of 'Potentilla', 'Festuca', 'Co-dominated'). Also it would be worth citing separately graph (A) (B) (C) (D) (E) (F) separately in the text instead of just 'Figure 2'.

In figures 1 and 2, it would help to write '(Observation Study)' after ...plant dominance gradient and '(Experimental Study)' after ...subordinate diversity and composition, respectively.

Experimental design

Good

Validity of the findings

Good

Additional comments

Thank you very much, you adequately answered my questions and integrated my comments and I am happy with the revised version. I can be published after minor changes indicated above are taken care of. It is nice to see more experiments on subordinate versus dominant species in grasslands and the interest in this topic.

---

## Round 0.3 · accepted · Accept

All minor suggestions have been adequately revised and I am happy with the revised version.

#